# Development of a Contact Force Model Suited for Spherical Contact Event

**Siyuan Wang** [1] **and Peng Gao** [2,*]

1   IT Faculty, Monash University, Melbourne, VC 3168, Australia
2   School of Mechatronical Engineering, Beijing Institute of Technology, Beijing 100081, China
*   Correspondence: gaopeng@bit.edu.cn

**Abstract:** The stiffness coefficient suited for a spherical contact body is developed by means of a contact semi-angle based on Steuermann's theory. The new static contact force model is close to the results of FEM when the index of the polynomial is equal to 2. The strain energy is derived according to the contact stiffness coefficient. Taylor expansion is used in the dissipated energy integration process to obtain a more accurate hysteresis damping factor. The new dynamic contact force model consists of the new stiffness coefficient and new hysteresis damping factor, which is suitable for the spherical-contact event with a high coefficient of restitution.

**Keywords:** new stiffness coefficient; new hysteresis damping factor; new contact force model; Steuermann's theory; FEM





## 1. Introduction

The spherical contact is a ubiquitous contact form in the contact phenomenon [1]. There is no energy to be dissipated when only the elastic force term is calculated [2]. This is because the original Hertz contact law produces the same path for the relationship be-tween the contact force and deformation during the loading and unloading process [3]. However, the energy dissipation has to be considered in the elastic contact phase. Alt-hough the damping force term produces the hysteresis damping loop representing the energy loss [4], the wave propagation is the primary reason for dissipating the kinetic energy when the elastoplastic or plastic deformation is not provoked during impact [5]. That is mainly because the internal damping of the contact material prevents wave prop-agation. According to the spring-damping model's formulation, both the elastic force term and damping force terms can be reconstructed. However, in the process of modifying the Hertz contact stiffness coefficient, some scholars neglected the relationship between the Hertz contact stiffness coefficient and the power exponent, resulting in the unit of the elastic force term being not Newton [6], which means that the coupling relationship be-tween the stiffness coefficient and power exponent cannot be treated independently [7], [8]. Otherwise, the developed contact force model violates the physical meaning of me-chanics. On the other hand, some scholars made a considerable contribution to deriving the hysteresis damping factor in a more accurate manner [9,10]. At least 15 different hys-teresis damping factors are developed based on the Hertz contact law [4,11–16]. Numer-ous researchers have studied spherical-contact events, and their research helps to strengthen the understanding of the contact mechanism between spheres. Liu et al. [17] considered the Hertz contact theory's limitation in the case of occurring relatively large deformation in the contact area. They proposed an approximate contact model of the spherical joint with clearance. This model's effectiveness is validated based on the finite element results. Fang et al. [1] established a new universal approximate model for the normal contact between frictionless spherical surfaces using analytic and numerical meth-ods. Goodman and Keer [18] developed an approximate solution for the contact between an elastic sphere and a flexible cavity without limiting the small contact region in the clas-sical elasticity theory framework. Sun and



Hao [19] studied the socket and ball's contact and implemented a comparison between the FEM solution and Hertz contact law.

In general, four fundamental theories can be followed to develop the contact force model [20]: (i) Hertz contact law; (ii) Winkler elastic foundation theory; (iii) Persson theory; (iv) Steuermann's theory. The first and fourth laws treated each body as an elastic half-space. For the Hertzian contact law, the Hertzian contact stiffness coefficient is the con-stant value when the nonlinear spring is proportional to the indentation depth powered by 3/2, which is independent of the contact deformation [21]. It is primarily suitable for a scenario that features a large clearance size and low load without friction [4,14]. The in-herent property of the Hertz contact law innately determines that all current contact force models must follow these assumptions [22]. The Winkler elastic foundation theory ig-nored the tangential force between contact bodies and treated the surface of the contact body as a series of springs that are independent of adjacent springs [23,24]. The contact pressure between the contact bodies is calculated using the spring's contact deformation and stiffness coefficient. Although the Winkler elastic foundation used a simple spring model to approximate the contact force, the spring's stiffness coefficient is hard to be de-termined. Therefore, the accuracy of the solution obtained using this theory is difficult to be guaranteed. The Persson theory [25] assumed to ignore the coupling relationship be-tween the radial and tangential displacement of the contact bodies. The functional rela-tionship between the maximum normal deformation and radial displacement must satisfy the rigid body's contact condition. Subsequently, it uses the geometrical constraint equa-tions to formulate the contact event between the contact bodies, which can provide a final solution to the impact between the clearance joint elements. However, this contact law is hard to be implemented. Steuermann's theory [18,26] used an axisymmetrical even-order polynomial to describe the contact pressure distribution. However, it is very strongly de-pendent on the polynomial index that depicts the profiles of the contact bodies. If the in-appropriate index of the polynomial is selected, the error will be enlarged. On the contrary, Steuermann's theory is the best choice to study the spherical-contact event [2,27] if the profiles of the contact bodies can be accurately described using the appropriate index of a polynomial [1,17]. That is why we choose Steuermann's theory to develop a contact force model for spherical-contact behavior.

This investigation formulated the new contact stiffness based on Steuermann's the-ory and proposed a new hysteresis damping factor based on energy conservation during impact. The index of the polynomial is determined as 2 by comparing it to the FEM model. Finally, a new contact force model tailored for the spherical contact bodies based on the spring-damping model is proposed. Compared to the existing contact force model, the proposed contact force model is more accurate in simulating impact behavior. More im-portantly, the related parameters in the new contact force model can be regulated accord-ing to the realistic engineering to approach the realistic contact profiles.

*Structure of this Investigation*

The structure of this investigation can be organized as follows: In Section 2, a new contact stiffness coefficient is proposed based on Steuermann's theory. In Section 3, a new hysteresis damping factor and a new static contact force model are proposed, which is validated by FEM. In Section 4, the overall performance of the new contact force model is systematically studied. The main conclusions are summarized in Section 5.

## 2. Pressure Distribution of Spherical Joint with Clearance

### 2.1. Hertz Contact Law

The Hertz law describes the contact pressure distribution between the contact bodies, which is developed by the contact event between two spheres. Accordingly, the Hertz law [19] must be subject to the following assumptions: (i) the material property of the contact bodies are linear elastic; (ii) the deformation should happen in the elastic range; (iii) the effect of surface shear and friction is ignored; (iv) the contact surfaces are contin-

uous and non-conforming; (v) each contact body is treated as an elastic half-space. The geometrical relationship between the spherical joint elements is an axisymmetrical problem, as dis-played in Figure 1. The contact pressure distribution can be written as

$$p(r) = p_0 \left(1 - \frac{r^2}{a^2}\right)^m \tag{1}$$

where $r$ is the projected horizontal distance between the points on the surface and the symmetry axis; $a$ is the radius of the contact area; $m$ is the pressure distribution exponent. The Hertz law assumes $m = 1/2$. $P_0$ is the maximum contact pressure $p_0 = \frac{3P_h}{2\pi a^2}$, $P_h$ is the contact force [28].

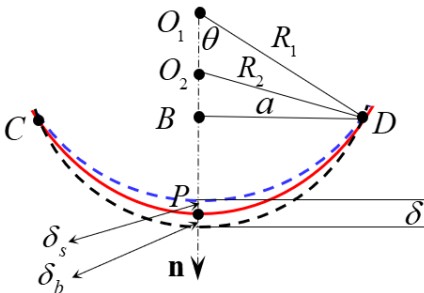

**Figure 1.** Geometrical relationship between socket and ball after contact.

The total normal deformation $\delta$ is given by,

$$\delta = \left(\frac{9P_h^2}{16RE^{*2}}\right)^{1/3} \tag{2}$$

where $\frac{1}{R} = \frac{1}{R_2} - \frac{1}{R_1}, \frac{1}{E^*} = \frac{1-v_1^2}{E_1} + \frac{1-v_2^2}{E_2}$, $E_1$, $E_2$ are Young's modulus; $v_1$ and $v_2$ are the Poisson's ratios.

Based on Equation (2), the contact force calculated based on Hertz law is written as,

$$P_h = \frac{4\delta E^*}{3}\sqrt{\delta R} \tag{3}$$

*2.2. Steuermann's Theory*

Steuermann's theory considered the contact pressure distribution that can be described by an axisymmetrical even-order polynomial with the form $A_n r^{2n}$, $n$ is the index of the polynomial to describe the profiles of the contact bodies. The contact pressure based on Steuermann's theory is written as [28],

$$p_n(r) = \frac{nA_nE^*a^{2n-2}}{\pi}\left\{\frac{2 \times 4 \ldots 2n}{1 \times 3 \ldots (2n-1)}\right\}^2 \times \left\{\left(\frac{r}{a}\right)^{2n-2} + \frac{1}{2}\left(\frac{r}{a}\right)^{2n-4} + \ldots + \frac{1 \times 3 \ldots (2n-3)}{2 \times 4 \ldots (2n-2)}\right\} \times \left(a^2 - r^2\right)^{1/2} \tag{4}$$

When the contact profiles are smooth and continuous, the contact force is given by [28],

$$P = \frac{4nA_nE^*a^{2n+1}}{2n+1}\frac{2 \times 4 \ldots 2n}{1 \times 3 \ldots (2n-1)} \tag{5}$$

In this case, the contact deformation can be written as,

$$\delta = \frac{2 \times 4 \ldots 2n}{1 \times 3 \ldots (2n-1)}A_na^{2n} \tag{6}$$

Therefore, the contact force [28] is written as,

$$P = \frac{4nE^*a}{2n+1}\delta \tag{7}$$

When $n$ approaches infinite, the profile of the contact body is approximated to a plane, and the point with the greatest contact stress is gradually away from the center point, which leads to the contact pressure distribution being infinite at the edges [17]. Moreover, it is worth noting that the radius of the contact area plays a crucial role in calculating the contact force based on Steuermann's theory.

*2.3. A new Contact Stiffness Coefficient*

A new contact semi-angle is proposed to estimate the contact radius between the socket and the ball. As shown in Figure 1, the socket and ball radii are $R_1$ and $R_2$, and the geometrical center of the socket and ball are $O_1$ and $O_2$, respectively. The total elastic con-tact deformation is assumed as $\delta$. In the entire contact process, the socket's deformation is treated as $\delta_s$, and the ball's deformation is assumed as $\delta_b$. Accordingly, the total elas-tic contact deformation is equal to the socket and ball's deformation $\delta = \delta_b + \delta_s$. The distance from the center of the socket and ball to the edge of the contact area is not the original radius because the elastic contact event happens. In Figure 1, the red curve is the edge of the socket and ball after contact. The distance from the center $O_1$ of the socket to the red edge is larger than $R_1$, which is equal to $R_1 + \delta_s$. Likewise, the distance from the center $O_2$ of the ball to the red edge is smaller than $R_2$, which is equal to $R_2 - \delta_b$. This geo-metrical relationship between the ball and socket establishes under the assumption of the small deformation. The edge contact points from the ball and socket are considered coin-cident before and after contact deformation, which are represented by point $C$ and point $D$. The eccentric distance between the socket and ball $O_1O_2$ can be calculated as $(R_1 + \delta_s) - (R_2 - \delta_b) = \Delta R + \delta$, ($\Delta R = R_1 - R_2$). According to the geometrical relation-ship, as shown in Figure 1, a new contact semi-angle $\theta$ can be defined based on the cosine law, which can be written as [29–31],

$$\theta = \arccos\frac{(\Delta R + \delta)^2 + R_1^2 - R_2^2}{2(\Delta R + \delta)R_1} \tag{8}$$

The radius of the contact area can be expressed as,

$$a = R_1 \sin\theta \tag{9}$$

Combining Equations (7) and (9), the contact force derived by Steuermann's theory can be rewritten as,

$$P = \frac{4nE^*R_1\sin\theta}{2n+1}\delta \tag{10}$$

Therefore, the new contact stiffness can be extracted from Equation (10) as,

$$K = \frac{4nE^*R_1\sin\theta}{2n+1}, \text{ when } n \to \infty, K_\infty = 2E^*R_1\sin\theta \tag{11}$$

This new contact stiffness coefficient includes the geometrical contact parameters, material property, and the profiles of contact bodies, which depends on the contact defor-mation rather than a constant value. This new contact stiffness coefficient is proposed based on Steuermann's theory, which is inspired by this literature [17] (the index of the polynomial is equal to 2). In order to validate the effectiveness of this new contact stiffness coefficient, a comparison analysis between the new contact stiffness coefficient and the contact stiffness coefficient developed by Caishan Liu should be implemented [17]. The contact semi-angle $\theta' = \arccos[\Delta R/(\Delta R + \delta)]$. The contact stiffness coefficient in the spherical joint with clearance is developed based on Steuermann's theory, which is ex-pressed as,

$$K_c = \frac{4nE^*R_1}{2n+1}\sqrt{1-\left(\frac{\Delta R}{\Delta R + \delta}\right)^2} \qquad (12)$$

This contact stiffness model is called as Liu model.

However, the Hertz contact stiffness coefficient can be extracted from Equation (3),

$$K_h = \frac{4E^*\sqrt{R}}{3}, R = \frac{R_1 R_2}{R_1 - R_2} \qquad (13)$$

It should be noted that the unit of the Hertz contact stiffness coefficient is N/m$^{3/2}$ ra-ther than N/m. Since the Hertz contact stiffness coefficient is independent of the contact deformation, which is not included in this comparative analysis. The contact parameters of the spherical joint with clearance are assumed as in Table 1.

**Table 1.** Structure parameters of the spherical joint with clearance.

| Elements | Young's Modulus (Pa) | Poisson Ratio | Mass (kg) | Radius |
|---|---|---|---|---|
| Socket | $2.068 \times 10^{11}$ | 0.29 | —— | $5.01 \times 10^{-2}$, $5.03 \times 10^{-2}$, $5.05 \times 10^{-2}$, $5.07 \times 10^{-2}$, $5.09 \times 10^{-2}$, $5.10 \times 10^{-2}$ |
| Ball | $2.068 \times 10^{11}$ | 0.29 | 0.02 | $5 \times 10^{-2}$ |

### 2.4. Distribution of the Contact Force for Pure Elastic Impact

To determine the value of index of the polynomial, an accurate finite element model for the contact between the ball and socket is established by ANSYS 18.1, as shown in Figure 2. The 6-node hexahedral elements are adopted, whose size is globally set as $4 \times 10^{-3}$ m. The meshes are comprised of 54,927 elements in total. When the clearance size is assumed as 0.5 mm, the distribution of the new model's contact force is almost identical to the Liu model no matter what the value of the index of the polynomial, as displayed in Figure 3.

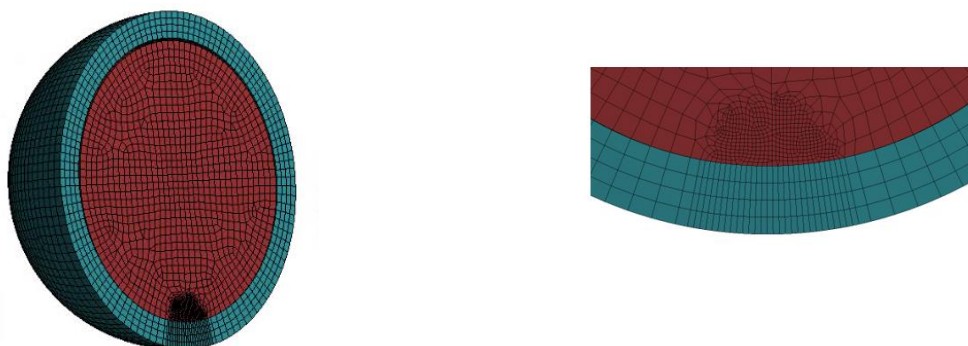

**Figure 2.** Finite element model of the spherical joint with clearance.

Subsequently, a comparison analysis between FEM and the new model is implemented, as displayed in Figure 3. The simulation results show that the contact force from the new model is close to the FEM when the index of the polynomial is equal to 2. This conclusion is consistent with the literature [17].

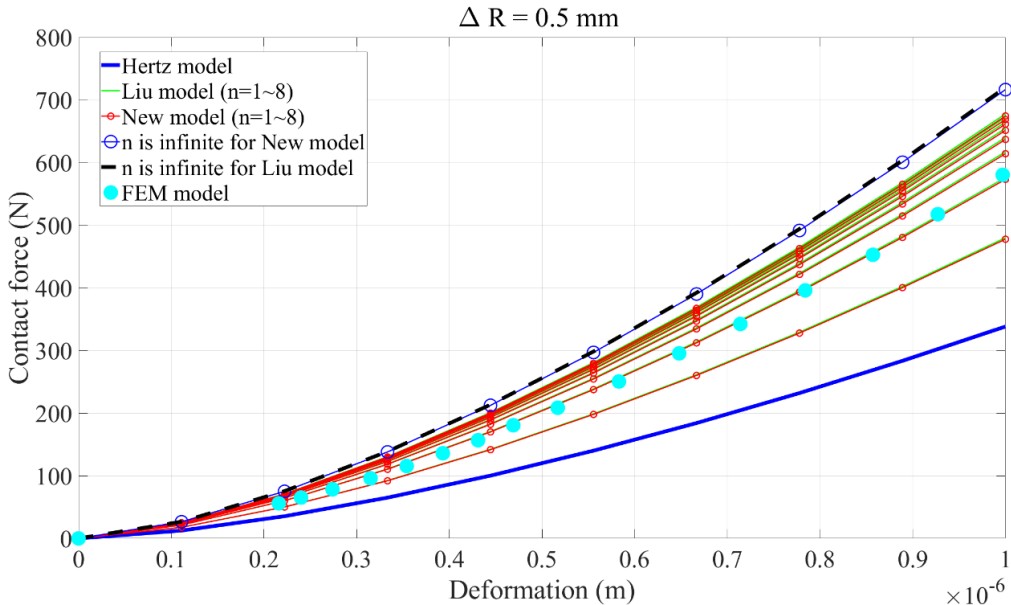

**Figure 3.** The proposed contact force model is validated by FEM.

When the clearance size is different in the spherical joint, a comparison analysis be-tween the Hertz model, Liu model, and the new model is implemented. Moreover, the contact force increases with the increasing contact deformation but reduces with the de-creasing clearance size. The error between the Liu model and the new model is small, which increases with contact deformation. The conclusion proves the correctness of the new elastic contact force model.

## 3. A New Contact Force Model

The kinetic energy of the contact bodies after contact is divided into three parts, in-cluding the kinetic energy of the contact bodies moving with the same common velocity, elastic strain energy, and dissipated energy due to internal damping.

### 3.1. Elastic Strain Energy during Contact

Based on Steuermann's theory, at the end of the compression phase, the elastic strain energy absorbed can be expressed according to Equation (11),

$$U^{(\max)} = \int_0^{\delta_{\max}} K\delta d\delta \tag{14}$$

The stored elastic strain energy can be integrated as,

$$U^{(\max)} = \int_0^{\delta_{\max}} \frac{4nE^*R_1\sin\theta}{2n+1}\delta d\delta = \frac{4nE^*R_1}{2n+1}\int_0^{\delta_{\max}}\sin\theta\delta d\delta = \frac{4nE^*R_1}{2n+1}\int_0^{\delta_{\max}}\sqrt{1-\cos^2\theta}\,\delta d\delta \tag{15}$$

Using Equation (8), the term $\int_0^{\delta_{\max}}\sqrt{1-\cos^2\theta}\,\delta d\delta$ is expressed as,

$$\int_0^{\delta_{\max}}\sqrt{1-\cos^2\theta}\,\delta d\delta$$
$$= \int_0^{\delta_{\max}}\sqrt{1-\left[\frac{(\Delta R+\delta)^2+R_1^2-R_2^2}{2(\Delta R+\delta)R_1}\right]^2}\,\delta d\delta \tag{16}$$
$$= \int_0^{\delta_{\max}}\frac{\sqrt{8R_1^2R_2\delta^3-4R_1^2\delta^4-8R_1R_2^2\delta^3+12R_1R_2\delta^4-4R_1\delta^5-4R_2^2\delta^4+4R_2\delta^5-\delta^6}}{2(R_1-R_2+\delta)R_1}\,d\delta$$

The elastic deformation is minimal. Its high order can be neglected, Equation (16) is rewritten as,

$$\int_0^{\delta_{max}} \sqrt{1-\cos^2\theta}\,\delta d\delta \approx \int_0^{\delta_{max}} \frac{\sqrt{8\left(R_1^2 R_2\delta^3 - R_1 R_2^2\delta^3\right)}}{2(R_1-R_2+\delta)R_1}d\delta$$

$$\approx \frac{\sqrt{2R_1 R_2\Delta R}}{R_1}\left[\frac{2}{3}\delta_{max}^{\frac{3}{2}} - 2\Delta R\delta_{max}^{\frac{1}{2}} + 2\Delta R^{\frac{3}{2}}\left(\frac{1}{\Delta R^{1/2}}\delta_{max}^{\frac{1}{2}} - \frac{1}{3\Delta R^{3/2}}\delta_{max}^{\frac{3}{2}} + \frac{1}{5\Delta R^{5/2}}\delta_{max}^{\frac{5}{2}} + O\left(\delta_{max}^{\frac{7}{2}}\right)\right)\right] \quad (17)$$

$$= \frac{2\sqrt{2}\sqrt{R_1 R_2}}{5\sqrt{\Delta R}R_1}\delta_{max}^{\frac{5}{2}}$$

Finally, the stored elastic strain energy in Equation (15) is given by,

$$U^{(max)} = \int_0^{\delta_{max}} \frac{4nE^*R_1\sin\theta}{2n+1}\delta d\delta$$

$$= \frac{4nE^*R_1}{2n+1}\cdot\frac{2\sqrt{2}\sqrt{R_1 R_2}}{5\sqrt{\Delta R}R_1}\delta_{max}^{\frac{5}{2}} = H\delta_{max}^{\frac{5}{2}} \quad (18)$$

where $H = \frac{8\sqrt{2}nE^*\sqrt{R_1 R_2}}{5\sqrt{\Delta R}(2n+1)}$, when $n\to\infty$, $H_\infty = \frac{4\sqrt{2}E^*\sqrt{R_1 R_2}}{5\sqrt{\Delta R}}$.

However, the elastic strain energy caused by the Hertz stiffness coefficient is expressed as,

$$U_{Hertz}^{(max)} = \frac{2}{5}K_h\delta_{max}^{\frac{5}{2}} \quad (19)$$

In Figure 4, the elastic strain energy increases with the increasing contact deformation. The changing trend of new strain energy is consistent with the Hertz model. Nevertheless, the magnitude of elastic strain energy calculated using Equation (19) is smaller than the new strain energy using Equation (18) no matter what the value of the index of the poly-nomial. Since the coefficient of the strain energy in Equation (18) is equal $\frac{8\sqrt{2}}{15}$ when $n$ is equal to 1, which is larger than $\frac{8}{15}$ in Equation (19).

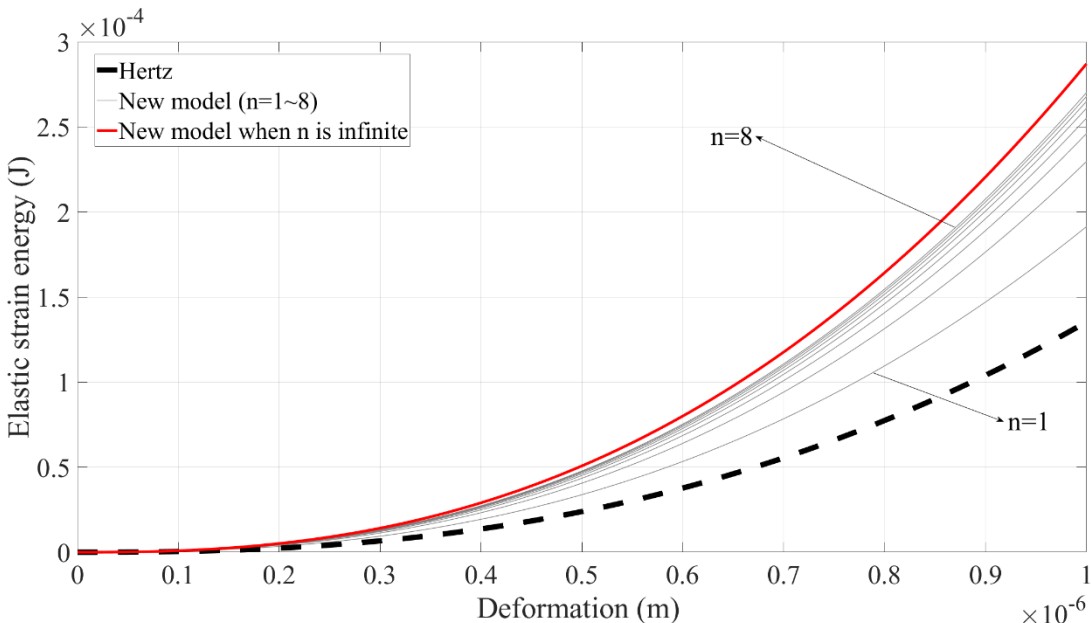

**Figure 4.** Comparison analysis of elastic strain energy between the Hertz model and the new contact force model.

### 3.2. A New Hysteresis Damping Factor

The dissipated energy through the work carried out by the damping force can be written as [16],

$$\Delta E = \oint \chi\delta^{\frac{3}{2}}\dot{\delta}d\delta \quad (20)$$

where $\chi$ is the hysteresis damping factor, $\dot{\delta}$ is the contact velocity.

The whole contact process contains two phases that are the compression and restitution phases. The entire dissipated energy caused by the damping factor can be given by [4],

$$
\begin{cases}
\Delta E_c = \chi \dot{\delta}^{(-)} \int_0^{\delta_{\max}} \delta^{\frac{3}{2}} \sqrt{1 - \left(\frac{\delta}{\delta_{\max}}\right)^2} d\delta \\
\Delta E_r = \chi \left|\dot{\delta}^{(+)}\right| \int_0^{\delta_{\max}} \delta^{\frac{3}{2}} \sqrt{1 - \left(\frac{\delta}{\delta_{\max}}\right)^2} d\delta
\end{cases}
\tag{21}
$$

where '*c*' represents the compression phase, and '*r*' represents the restitution phase. $\Delta E_c$ and $\Delta E_r$ are the dissipated energy in the compression and restitution phases. $\dot{\delta}^{(-)}$ and $\dot{\delta}^{(+)}$ are the relative compression velocity and the relative separating velocity, respectively.

The integral process for the whole contact process can be obtained as,

$$
\Delta E = \Delta E_c + \Delta E_r = \chi \left(\dot{\delta}^{(-)} + \left|\dot{\delta}^{(+)}\right|\right) \delta_{\max}^{\frac{5}{2}} \int_0^1 x^{\frac{3}{2}} \sqrt{1 - x^2} dx \approx \frac{13}{50}(1 + c_r) \chi \dot{\delta}^{(-)} \delta_{\max}^{\frac{5}{2}}
\tag{22}
$$

The coefficient of restitution can be defined as $c_r = \frac{\left|\dot{\delta}^{(+)}\right|}{\dot{\delta}^{(-)}}$.

The dissipated energy at the end of the compression phase can be written as,

$$
\Delta E_c = \chi \dot{\delta}^{(-)} \delta_{\max}^{\frac{5}{2}} \int_0^1 x^{\frac{3}{2}} \sqrt{1 - x^2} dx \approx \frac{13}{50} \chi \dot{\delta}^{(-)} \delta_{\max}^{\frac{5}{2}}
\tag{23}
$$

At the end of the compression phase, the initial kinetic energy $T^{(-)}$ of the contact bodies has three different destinations, including the kinetic energy $T^{(\max)}$ of the system at the end of the compression phase, the maximum elastic strain energy $U^{(\max)}$ stored in the contact bodies, and dissipated energy $\Delta E_c$ in the compression phase without resti-tution phase.

$$
T^{(-)} = T^{(\max)} + U^{(\max)} + \Delta E_c
\tag{24}
$$

Therefore, Equation (24) can be rewritten as,

$$
\frac{1}{2} m_i v_{0i}^2 + \frac{1}{2} m_j v_{0j}^2 = \frac{1}{2}(m_i + m_j) v_{ij}^2 + H \delta_{\max}^{\frac{5}{2}} + \frac{13}{50} \chi \dot{\delta}^{(-)} \delta_{\max}^{\frac{5}{2}}
\tag{25}
$$

where $m_i$ and $m_j$ are the mass of the contact bodies, $v_{0i}$ and $v_{0j}$ are the pre-impact velocities. $v_{ij}$ is the common post-impact velocity.

Equation (25) can be simplified based on the linear momentum balance,

$$
\delta_{\max}^{\frac{5}{2}} = \frac{25m \left(\dot{\delta}^{(-)}\right)^2}{50H + 13\chi \dot{\delta}^{(-)}}
\tag{26}
$$

where $m = \frac{m_i m_j}{m_i + m_j}$.

From the coefficient of restitution point of view, the dissipated energy can be expressed according to the energy balance law [4],

$$
\Delta E = \frac{1}{2} m \left(\dot{\delta}^{(-)}\right)^2 (1 - c_r^2)
\tag{27}
$$

Combine Equations (22), (26) and (27) to obtain a new hysteresis damping factor,

$$
\chi = \frac{50H(1 - c_r)}{13\dot{\delta}^{(-)} c_r}
\tag{28}
$$

Combining Equations (11) and (28), a new contact force model suited for the spherical-contact event can be formulated based on Steuermann's theory,

$$F_N = K\delta + \frac{50H(1-c_r)\delta^{\frac{3}{2}}}{13c_r}\frac{\dot{\delta}}{\dot{\delta}^{(-)}}, K = \frac{4nE^*R_1\sin\theta}{2n+1} \tag{29}$$

This contact force model is referred to as the new contact force model.

Compared to the new contact force model, the other two famous contact force models, including Lankarani-Nikravesh [32] and Flores et al. contact force models [4] are introduced in this section. The Lankarani-Nikravesh contact force model is expressed as,

$$F_{L-N} = K_h\delta^m\left[1 + \frac{3(1-c_r^2)}{4}\frac{\dot{\delta}}{\dot{\delta}^{(-)}}\right] \tag{30}$$

The Flores et al. contact force model is given by,

$$F_f = K_h\delta^m\left[1 + \frac{8(1-c_r)}{5c_r}\frac{\dot{\delta}}{\dot{\delta}^{(-)}}\right] \tag{31}$$

As shown in Figure 5, the magnitude of new hysteresis damping factor decreases with the increasing coefficient of restitution. The relatively large coefficient of restitution represents that only a small amount of kinetic energy is to be dissipated during contact. However, the magnitude of the new hysteresis damping factor is larger than the other contact force models, whatever the value of the index of the polynomial. Although its change trend is similar to the Gonthier et al. [12], Flores et al., and Hu and Guo models [33], its magnitude is conspicuously larger. The primary reason lies in that the form of the new hysteresis damping factor in Equation (28) is similar to the Gonthier et al., Flores et al. and Hu and Guo models' denominator includes the square of the coefficient of restitution rather than one power. In addition, the coefficient of the new hysteresis damping factor is larger than the other three contact force models.

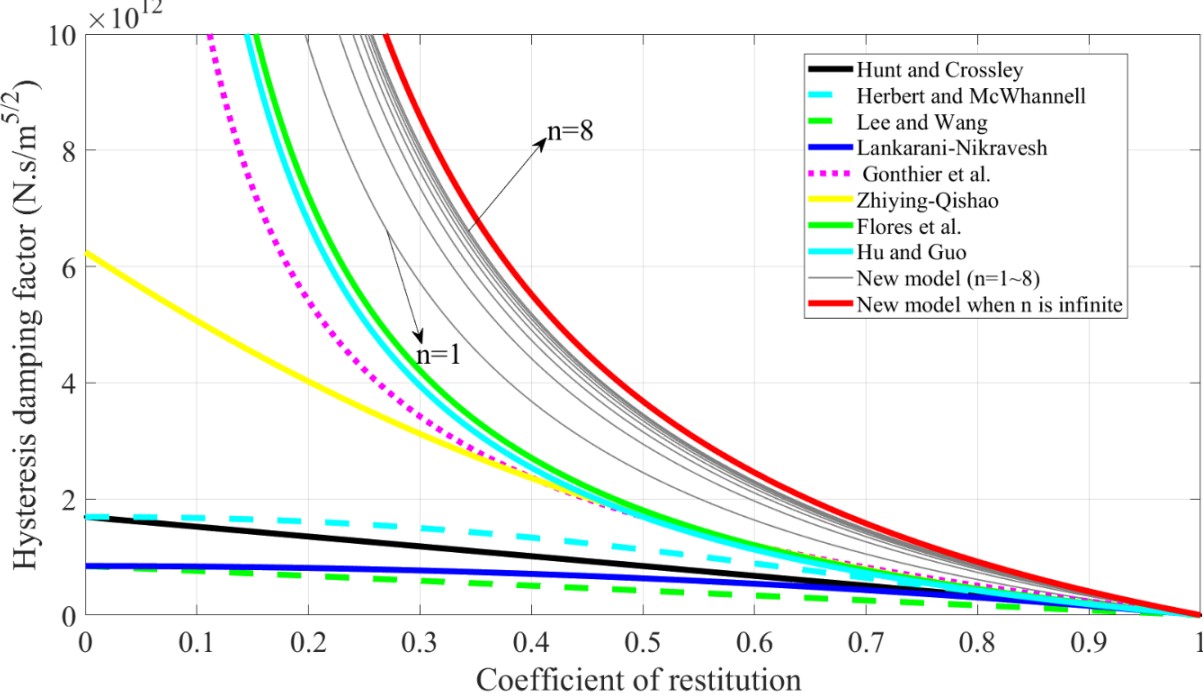

**Figure 5.** Comparison analysis of hysteresis damping factors.

## 4. The Dynamic Performance of the New Contact Force Model

In order to test the dynamic performance of the new contact force model, a spherical joint with clearance is taken as an example, which can be seen in Figure 6. In this model, the socket is fixed on the ground. The ball has an initial velocity that is equal to 0.3 m/s to impact the socket. The mass of the ball is assumed as 1 kg. The integral parameters are determined in Table 2.

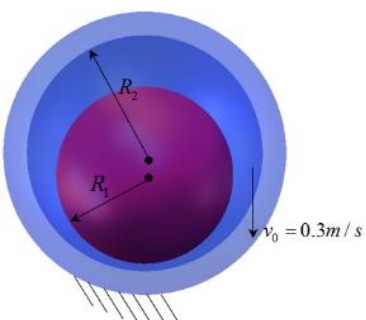

**Figure 6.** Contact model between socket and ball.

**Table 2.** Integral parameters.

| Parameters | Parameter Values |
| --- | --- |
| Integrator | Ode45 |
| Relative error | $1 \times 10^{-9}$ |
| Absolute error | $1 \times 10^{-9}$ |
| Initial time step | $1 \times 10^{-6}$ |
| Time span | $1 \times 10^{-4}$ |

### 4.1. Determination of the Coefficient of Restitution

Before carrying out the numerical simulation, the coefficient of restitution is an index to evaluate the energy dissipation during the contact-impact event [3]. How to determine its value is of important influence for the calculation precision? From a physical point of view, if the discrepancy between the post-and pre-restitution coefficient is significantly smaller so as to be ignored [34], the contact process described by the contact force model is more approximated to the actual situation [35]. In order to determine the pre-restitution coefficient [5], an error percentage is provided for estimating the difference between the post-and pre-restitution coefficient, which is given by,

$$Error = \frac{\left| c_r - \left| v^{(+)}/v^{(-)} \right| \right|}{c_r} \times 100\% \tag{32}$$

where $v^{(+)}/v^{(-)}$ is the post-restitution coefficient. $v^{(+)}$ is the post-velocity.

Since this new contact force model is applicable to elastic deformation, when the co-efficient of restitution is small, the error is very large. On the contrary, the error is less than 2.5% when the coefficient of restitution is larger than 0.9 no matter what n takes a value from 1 to 8 in Figure 7. This conclusion accounts for that the new contact force model is suited for the high value of the coefficient of restitution. Therefore, in order to make the simulation results close to the real contact situation, the pre-restitution coefficient is determined as 0.9. Once the coefficient of restitution is determined, the energy dissipation can be calculated by integrating the damping term in Equation (20). In Figure 8, the mag-nitude of the energy loss increases with the increasing contact deformation regarding these contact force models. As for the new contact force model, its magnitude keeps con-sistent with the other contact force models, which increases with the increasing index of the polynomial.

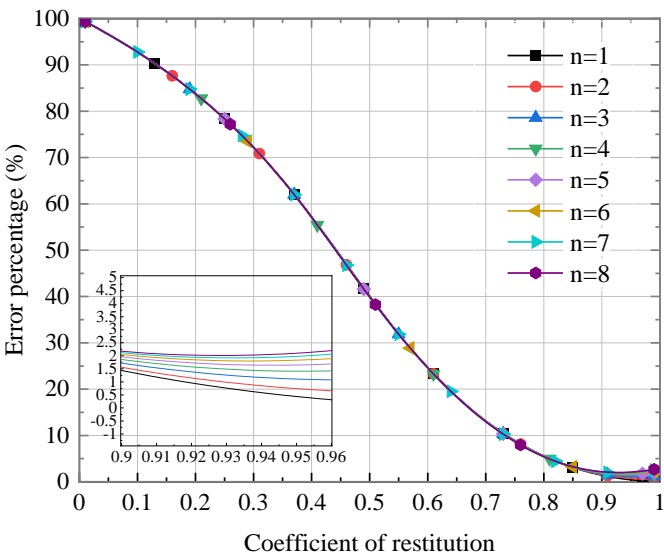

**Figure 7.** Evaluation of the coefficient of restitution.

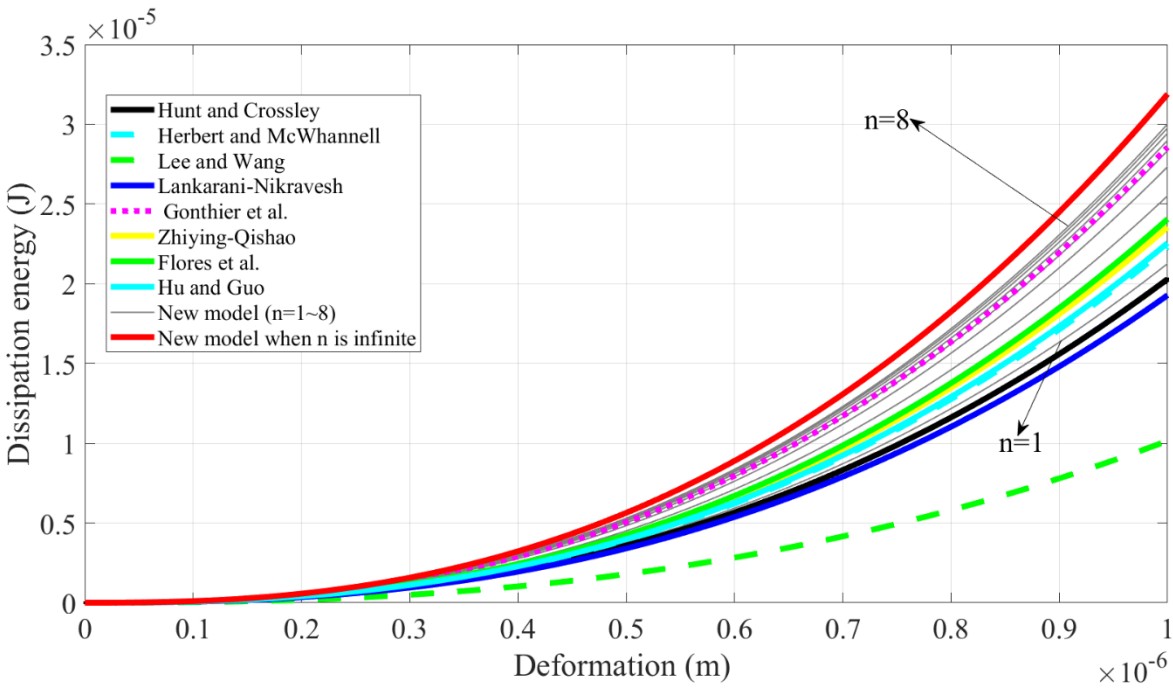

**Figure 8.** Comparison analysis of the energy dissipation.

Since the magnitude of the new hysteresis damping factor is larger than the other contact force models in Figure 5, the stored elastic strain energy in Figure 4 is also greater than the other contact force model. The dissipated energy during the contact process in Figure 8 can keep consistent with the other contact model, which illustrates the new hys-teresis damping factor is capable of accurately describing the dissipated energy. The initial kinetic energy has three different destinations after impact, wherein the post-kinetic en-ergy is large, the stored strain energy is moderate, and the dissipated energy is small.

*4.2. Dynamics Simulation*

In the case of the same initial contact condition, the maximum penetration depth from the new contact force model is achieved in a short time compared to the Hertz model, L-N model, and Flores et al. models in Figure 9a. That is mainly because the most kinetic energy is stored in the contact body as strain energy in the same contact periodic. Moreo-ver, the

magnitude of the contact force is the largest as well, whatever the value of the index of the polynomial. What reason leads to a significant discrepancy?

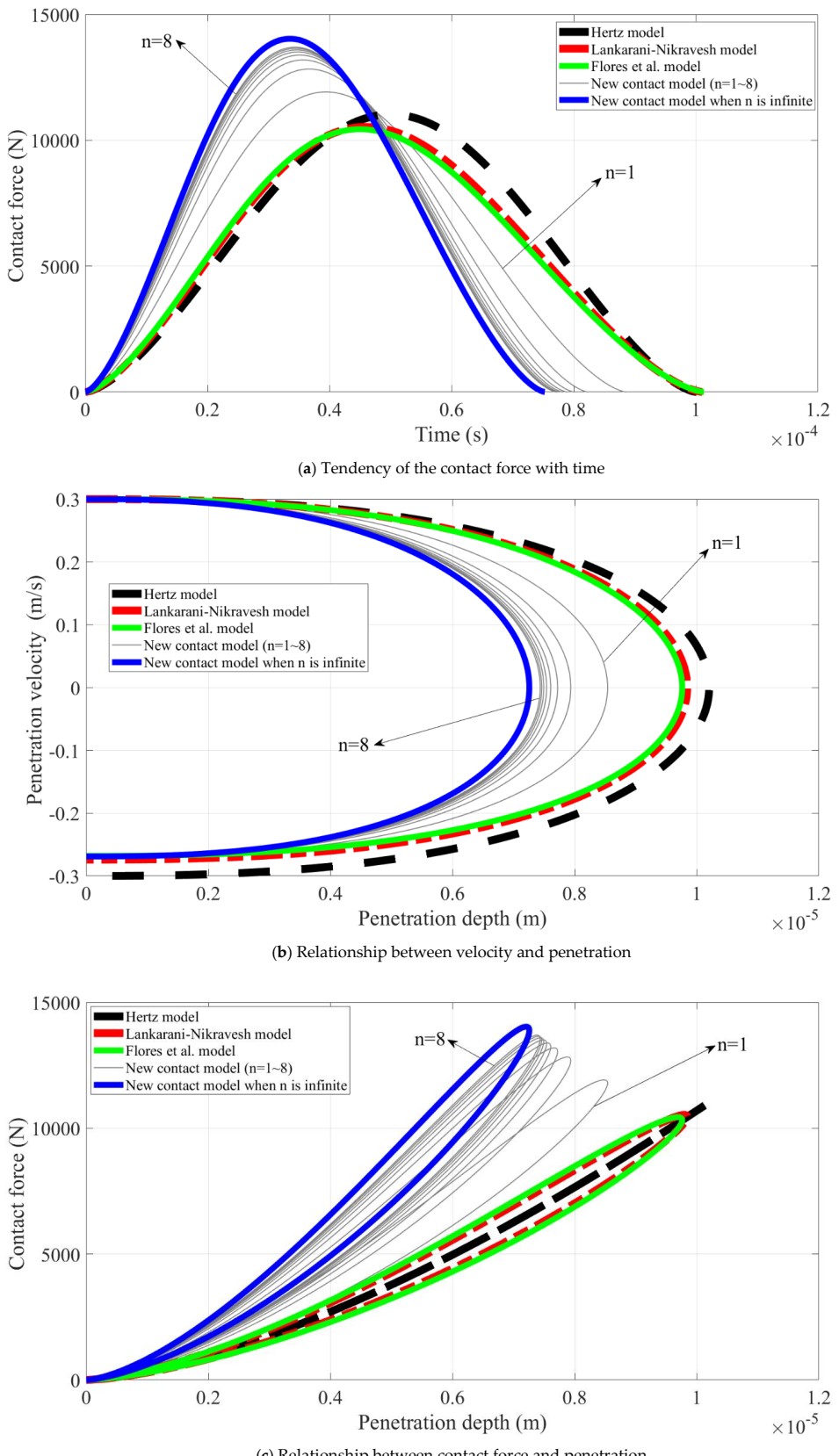

(**a**) Tendency of the contact force with time

(**b**) Relationship between velocity and penetration

(**c**) Relationship between contact force and penetration

**Figure 9.** Comparison analysis between the existing contact models and the proposed contact model.

There are two main reasons to illustrate this phenomenon. On the one hand, the damping force term in the new contact force model is the largest value of the other models. On the other hand, although the Hertz contact stiffness coefficient is significantly larger than a new model, the power exponent in the elastic force term is 1 rather than 3/2 in the new contact force model, unlike its value is equal to 3/2 to the Hertz model, L-N model, and Flores models. Comprehensively, the magnitude of the contact force from the new contact force model is the largest. The new contact force model needs the least time to finish the contact event. Hence, the penetration velocity reduced most sharply, as dis-played in Figure 9b. The relationship between the contact force and penetration depth is implemented in Figure 9c. Clearly, the area of the hysteresis damping loop from the new contact force model is relatively larger than the L-N model and Flores et al. models, which accords with the results in Figure 5. The amount of energy dissipation from the new con-tact force model is the same whatever the value of the index of the polynomial, which illuminates the energy loss of the new contact force model is independent of the index of the polynomial. The entire dynamic contact process between the ball and socket can be successfully described using the new contact force model. Only the contact force and con-tact duration are different from the L-N and Floes et al. model, which validates the new contact force model's reasonability.

## 5. Conclusions

In light of Steuermann's theory can describe the geometry shape and contact pressure distribution by an axisymmetrical even-order polynomial, this investigation proposes a novel contact force model tailored for the spherical-contact event. This contact force model is composed of a new contact stiffness coefficient and a new hysteresis damping factor.

In the process of designing the static contact model, a contact semi-angle is redefined according to the geometrical relationship between the socket and ball. Subsequently, the contact stiffness coefficient is formulated using this new contact semi-angle and axisymmetrical even-order polynomial. The simulation results show that the new static contact force model keeps consistent with the Liu model no matter what the value of the index of the polynomial. In order to confirm this important parameter, the same contact model is established using FEM, which proves the index of the polynomial should be equal to 2. This conclusion accords with the literature.

Due to that the new contact stiffness coefficient is the function of contact deformation, and the stored strain energy is no longer $\frac{2}{5}K_h\delta_{\max}^{\frac{5}{2}}$ [4]. Moreover, according to the law of energy conservation during impact, a new hysteretic damping factor is proposed, which is relatively large in magnitude compared to other existing damping factors developed using the Hertzian contact law. Furthermore, the effect of the index of polynomial on the dynamic performance of the new contact force model is discussed. The results show that dynamic performance is sensitive when its value changes from 1 to 3. After that, the dy-namic characteristics are not sensitive to this parameter. The first three values represent the relatively simple geometrical shape, respectively. However, the influence of dynamic contact behavior is not significant when the contact body shape becomes complex.

Compared to the L-N model and Flores et al. model, the new contact force model is significantly different from them. On the one hand, the new stiffness coefficient depends on the contact deformation unlike the Hertz contact stiffness coefficient. The power exponent of deformation in the elastic force term is 1 rather than 3/2, which is determined by Steuermann's theory. On the other hand, the new hysteresis damping factor causes more energy to be dissipated. This conclusion illustrates that this model is suitable for the spher-ical-contact event with high restitution coefficient.

**Author Contributions:** Conceptualization, methodology, validation, writing—review and editing, S.W. and P.G.; writing—original draft preparation, software, data curation, visualization, S.W.; formal analysis, investigation, resources, supervision, P.G.; project administration, funding acquisition, P.G.; All authors have read and agreed to the published version of the manuscript.

**Funding:** This research received no external funding.

**Institutional Review Board Statement:** Not applicable.

**Informed Consent Statement:** Not applicable.

**Data Availability Statement:** Not applicable.

**Conflicts of Interest:** The author declares no potential conflict of interest concerning the research, authorship, and/or publication of this article.

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
