# Peer review of "Development of a Contact Force Model Suited for Spherical Contact Event"

_actuators, doi:10.3390/act12020089_

Round 1

Reviewer 1 Report

This paper proposed a new contact model based on Steuermann's theory, and applied it to the contact behavior between the spherical contact bodies. Its effectiveness is validated by the Finite Element Method. However, there are some issues needed to be clarified as follows: 

1. Rewrite the introduction, and remove the irrelative references;

2. Please highlight your contributions; why did you develop this contact force model when the FEM can handle this contact behavior. 

3. remove the unnecessary description regarding the known knowledge.

4. elucidate the merits of the proposed contact force model compared to the existing contact force model.

Author Response

Response 1: Thanks for your reminder, we have revised the introduction of this paper. The reference has also been updated according to your suggestions.

Response 2: The contributions of this paper have been restated at the end of the introduction. In addition, the novel contact force model is tailored to the impact event between the spherical contact bodies. The related parameters in the new contact force model can be regulated according to realistic engineering, more importantly, it can be applied to the dynamics simulation of multibody systems associated with large rigid motion and flexible deformation. However, the FEM is hard to be worked with the dynamic simulation with impact behavior, namely, it is difficult to be explored secondly on the basis of the actual engineering application. That is why we develop this contact force model base on Steuermann's Theory.

Response 3: Thanks for your reminder, we have removed the unnecessary content to condense the length of the manuscript.

Response 4: The merits of this model can be seen at the end of the introduction and conclusion parts of this paper.

Reviewer 2 Report

The following questions are raised. 

1. In the introduction, so many references are listed, their differences and relations should be clarified. the authors should quote only these closely related works. 

2. on line 153, is n a constant?

3. whether does the pressure in eq. (5) meet the problem in figure 1? maybe it should be verified. 

4. in their new model, how to choose n, the author should give some method to determine this value, not just list n=1..8

Author Response

Response 1: Thanks for your insightful comments on this paper. The irrelated references have been removed. Only pertinent works are cited in this paper.

Response 2: Your comment on the parameter n is correct because this value is used to depict the profiles of the contact bodies. When n approaches infinite, the profile of the contact body is approximated to a plane.

Response 3: Eq. (5) is derived when the contact profiles are smooth and continuous. Further, the contact profiles of the contact bodies have been controlled by the index n of the polynomial. Therefore, this polynomial can describe the contact event like in Fig.1 which is a local magnified schematic diagram representing the analogous spherical contact event.

Response 4: Thanks for your suggestions. We have restated the selection of this parameter n. In general, this value can be chosen from 1 to 3 when the geometry of the contact body is simple. While the profiles of the contact bodies are complicated, this value is hard to be determined or is not sensitive to the contact behavior, that is, at this moment, this model will lose its advantage. We can consider using another contact model when the geometry of the contact bodies is complex. If the profile of the contact body infinitely approaches the spherical shape, it is best to select this value equal to 2.

Reviewer 3 Report

Dear authors,

Unfortunately, I must comment that the modeling you propose is not new and many of your studies are not useful at all. Moreover, the paper includes serious deficiencies. Some of them are listed below (more can be found in the attached review file):

·  Right at the beginning of the abstract, you claim that you have propose a “new contact semi-angle” to estimate the contact radius between the socket and ball. Section 2.3 (lines 170-190) deals with the derivation that results in Eq. (8). However, this result is by no means new and has been known for a long time. For example, Eq. (7) in the classical work of Bartel et al. [1]  from 1985 is identical to your supposedly new formula. This formula, which comes from the simple cosine theorem, has also been used in more recent work (see e.g. Askari et al. [2], Eq. (14)). In the work by Askari et al. [2] as well as Heß & Forsbach [3] the radial displacement is set up completely analogously via the cosine theorem.

· The above three papers are missing from the reference list. Your reference list contains many important papers, but unfortunately also errors. For example, in the first sentence of the introduction, you cite a one-page book-review by Keer of Johnson's famous book on contact mechanics to motivate the contact-impact event in practical applications. From line 110 on page 3, Persson's contact theory is discussed in more detail, citing a paper by Manners and Greenwood [48]. Manners and Greenwood actually discuss Persson's contact theory in detail. However, this theory has nothing at all to do with the problem at hand. In fact, the authors explain the famous theory developed by Bo Persson for contacts of surfaces with multiscale roughness. However, what you probably want to cite is certainly the theory developed by Allan Persson in his 1964 doctoral dissertation [4] on cylindrical elastic bodies in contact.

·  In Section 2.3, you define the contact stiffness coefficient of your “new” model and compare it to a relation given by Liu (your Ref. [23]). Several times you mention that the contact stiffness coefficients are very close to each other no matter what value of the index of the polynomial (e.g. at lines 219-222). This thesis is supported by figures 2 (a) to (c). However, the whole investigation is completely superfluous. Both models use Steuermann's theory, i.e. equation (10), and hence the polynomial coefficient cannot make a difference. Only for the sine of the contact half angle a different function of the indentation depth is used. While you use Eq. (8) from the cosine theorem, Liu apply an approximated relation that follows exactly from Eq. (8), assuming that the clearance and the indentation depth are much smaller than the radii of curvature of both bodies. These two conditions are fulfilled for almost conformal contacts and also for the data you provide in table 1. For radial displacement under these conditions, the same approximate transition can be found in [3] (compare Eqs. (5) and (6)). Thus, the differences in the stiffness coefficients arise only due to the kinematics, completely independent of the polynomial exponent. Figures 5 and 6 are thus also completely superfluous. The percentage deviation in the coefficient of stiffness according to Figure 2 must be identical to the percentage deviations in the contact force according to Figure 6. This is trivial. There is no other way because you only multiply by the indentation depth. Also, the statement in lines 227 to 229 that with increasing indentation depth and clearance, the error becomes larger is immediately clear when Liu's geometric assumptions (see above) are taken into account.

·     In lines 98 to 100 it is stated that

As for the Hertz contact law, the Hertz contact stiffness coefficient is a constant value when the contact material and the curvature radius of the contact bodies are determined, which is independent of the contact deformation.

This statement is only correct if you assume a nonlinear spring proportional to the indentation depth powered by 3/2. In lines 212 to 214 you even point out that the stiffness coefficient according to Hertz has a different unit of measurement than that according to e.g. Eq. (12). For this reason alone, you cannot compare both. To be able to compare both Eq. (13) must be multiplied by the square root of the indentation depth! 

·    Section 3.2 makes use of Hunt and Crossley’s model. After Eq. (21) (starting from Line 309) you denote quantities for the compression and restitution phases (“…where ‘c’ represents the compression phase, ‘r’ represents the restitution phase…”). However, the quantities of the restitution phase are never used in the following. It is said that in Eq. (23) the total dissipated energy is determined, but this is not correct. Instead, the calculation from Eqs. (21) or (22) for the dissipated energy in the compression phase is simply continued.

Due to the deficiencies mentioned above, the contact force given in equation (28) with alleged new stiffness coefficient and restitution parameter must be strongly questioned. The same applies to the comparison with well-established contact laws. 

References:

   [1]      Bartel, D. L., Burstein, A. H., Toda, M. D., & Edwards, D. L. (1985). The effect of conformity and plastic thickness on contact stresses in metal-backed plastic implants.

   [2]      Askari, E., & Andersen, M. S. (2018). A closed-form formulation for the conformal articulation of metal-on-polyethylene hip prostheses: Contact mechanics and sliding distance. Proceedings of the Institution of Mechanical Engineers, Part H: Journal of Engineering in Medicine, 232(12), 1196-1208.

   [3]      Heß, M., & Forsbach, F. (2021). An Analytical Model for Almost Conformal Spherical Contact Problems: Application to Total Hip Arthroplasty with UHMWPE Liner. Applied Sciences, 11(23), 11170.

   [4]      Persson, A. (1964). On the stress distribution of cylindrical elastic bodies in contact. Chalmers University of Technology.

Further comments can be found in the attached file!

Author Response

Thanks for your insightful comments. All responses to the comments point by point can be found in the attachment. 

Round 2

Reviewer 3 Report

Dear authors,

the first part of your revised manuscript now makes much more sense and the limitation to only essential literature contributions is an improvement. The same applies to the addition of the equations in Section 3.2. in the second part of the manuscript (following the work of Flores et al.), which makes the calculation path for the hysteresis damping factor more understandable. However, in addition to minor queries, there is still a need for clarification in the calculation of the hysteresis damping factor. At first glance, it seems that the hysteresis damping factor you calculated is not correct. This would have a strong impact on all further calculations and graphs you show. Therefore, it is very important that you address the points listed in the attached pdf-file!

Author Response

Response 1: Thanks for your insightful comment. It is impossible that the quality of the paper has significant improvement without your excellent suggestions. The hysteresis damping factor has a mistake for the power exponent of the coefficient of restitution. We have revised it according to your comments.

Response 2: Thanks for your message.

Response 3: Thanks for your reminder. We have imposed the related contents.

Response 4: We completely agree with your suggestion. We have revised it.

Response 5: Thanks for your careful review work, we have revised these typos.